# Study on Ultrathin Silver Film Transparent Electrodes Based on Aluminum Seed Layers with Different Structures

**DOI:** 10.3390/nano12193540

**Published:** 2022-10-10

**Authors:** Dong Li, Yongqiang Pan, Huan Liu, Yan Zhang, Zhiqi Zheng, Fengyi Zhang

**Affiliations:** School of Optoelectronic Engineering, Xi’an Technological University, Xi’an 710021, China

**Keywords:** ultra-thin silver film, infiltration layer, transparent electrode

## Abstract

Ag has the lowest electrical resistivity among all metals, and at the same time, the best optical properties in the visible and near-IR spectral range; it is therefore the most widely employed material for thin-metal-film-based transparent conductors. In this work, an ultra-thin transparent silver film electrode with aluminum as seed layer was prepared by a resistive thermal evaporation technique. Using a range of electrical, optical and surface morphology techniques, it can be noted that the presence of the thin layer of aluminum changes the growth kinetics (nucleation and evolution) of the thermal evaporation of Ag, leading to silver films with smooth surface morphology and high electrical conductivity, and the threshold thickness of the silver film is reduced. It is inferred that the aluminum layer showed a good infiltration effect on the ultra-thin silver film, by analyzing the transmittance spectrum, sheet resistance and surface morphology. Moreover, the average transmittance of silver film with 10 nm is 40% in the 400–2500 nm band, whereas the sheet resistance is 13 Ωsq ^−1^. A series of experiments show that the introduction of Al seed layer has certain effect on improving the properties of transparent conductive silver films. Then, a new method for deposition of 1 nm Al seed layer was proposed; that is, the 1 nm aluminum infiltrated layer is divided into two or more layers, and the average transmittance of silver film with 5 nm is 60% in the 400–2500 nm band, whereas the sheet resistance does not exceed 100 Ω sq^−^^1^.

## 1. Introduction

The ideal transparent electrode requires high optical transmittance and low sheet resistance. High flexibility is also a critical and indispensable component of emerging flexible optoelectronic devices. Flexible transparent electrodes are in significant demand in applications including solar cells [1], light-emitting diodes [2,3,4], curved surface screens, touch panels [5] and other optoelectronic devices. Indium tin oxide (ITO) is the conventional selection and most widely used for the transparent electrode because of its high visible transmittance and electrical conductivity. However, the low abundance of the indium element on earth is a limiting factor of this material. In addition, its applications in emerging flexible optoelectronic devices are significantly hindered by both the poor mechanical flexibility and the high annealing temperature needed to reduce its resistivity. Recently, several other transparent conductive materials have been developed to address these issues. For instance, doped metal oxides, thin metals, conducting polymers, and nanomaterials (including carbon nanotubes, graphene, and metal nanowires) [6], etc., have gradually become effective substitutes for ITO film. Metal thin film is an ideal material for transparent electrodes because of its simple preparation process, low cost, excellent mechanical flexibility and uniform photoelectric properties. In addition, compared with the ITO film, the transmittance of silver metal film is relatively poor, but it is featured with good conductivity and flexibility, and the silver resources are more abundant than the indium resource [7].

A much thinner Ag layer is necessary for higher transmittance, but it is well known that Ag grows in the Volmer–Weber mode on glass substrates [8]. The conductivity of silver films will increase with increased thickness, but the permeability of silver films will be affected if the thickness is too thick. Therefore, it is necessary to introduce a seed layer to reduce the threshold thickness of the film and change the growth mode of the silver film, so that the silver film can be continuous at a lower thickness.

Numerous metals from the periodic table have been used as a seed layer, such as Nickel(Ni) [9], Copper(Cu) [10], and Gold(Au) [11]. Stefaniuk et al. studied the effect of Ge, Ni and Ti as wetting layers on the resistivity of silver films [12]. Wang et al. provided an effective method for preparing transparent silver electrodes and found that a small amount of oxygen doping can improve the optical and electrical properties of silver films [1]. Logeeswaran et al. studied the effect of a metal Ge infiltration layer on the morphology of silver films on glass substrates by comparing silver films with and without a Ge infiltration layer [13]. Lv Jing et al. from Fujian Normal University studied the effect of copper and aluminum as infiltration layers of silver films on the thermal stability and resistivity of 20 nm silver films [14]. Through thermal evaporation technology and doping with different elements, Xue Weining from Zhejiang University successfully grew aluminum–silver-doped films with a smooth surface and good thermal stability [15]. At present, there are two main methods for infiltrating silver films with metals. One is to prepare silver films by doping other metals with silver [16], and the other is to infiltrate silver films directly as seed layers to reduce the threshold thickness (the thickness at which the films begin to grow continuously) of the silver films [17]. The doping ratio of silver-doped films is difficult to control, and the process is complicated. There are relatively few studies on the preparation of transparent conductive silver films directly using aluminum as the infiltration layer, and the existing studies on ultra-thin silver transparent conductive films mainly focus on the transparency and conductivity in the range of visible light band.

In this work, first, the thermal evaporation technique was used to prepare ultrathin silver transparent conductive film, and then, based on the silver film transmittance, sheet resistance and SEM image measurement and analysis, the effect of aluminum as a seed layer on the photoelectric properties of silver films on a K9 glass substrate was studied. The optimal infiltrating thickness of the aluminum layer was determined, and silver films with different thickness were prepared. The optical and electrical properties of ultrathin transparent silver conductive films in the 400–2500 nm band were studied.

## 2. Model and Experiment 

### 2.1. Model of Thin Metal Film Growth

Metal film growth on a substrate generally follows several stages in a sequence, namely nucleation, coalescence, and thickness growth [18]. Depending on the strength of surface energy of the substrate (γ_s_), surface energy of the metal (γ_m_), and metal/substrate interface energy (γ_m/s_), there are three nucleation modes [19]. (i) Frank–van der Merwe mode (layer by layer growth): layers of material grow on top of one another. The interaction between adjacent substrate atoms and metal adatoms is stronger than that between adjacent metal adatoms. (ii) Volmer–Weber mode (island growth): isolated 3D metallic islands form on the substrate. The interaction between adjacent metal adatoms is stronger than that between metal adatoms and substrate atoms. (iii) Stranski–Krastanov mode (layer plus island growth): one or two monolayers of material form first, followed by individual islands on top. This is a situation between (i) and (ii), and involves a change of interaction energies between these atoms. The surface energies of metal film deposition are shown in Figure 1. Young′s equation is satisfied between the surface energies at equilibrium [20] Equation (1).
(1)$$\gamma_{s}=\gamma_{m/s}+\gamma_{m}\cos\theta$$

When the interface between the film and the substrate is zero, the film growth is the ideal lamellar growth, in the island-like growth pattern θ > 0, indicating that γ_s_ < γ_m/s_ + γ_m_ [21]. The adhesion energy E_adh_ is the energy to separate the metal/substrate interface in a vacuum and can be expressed as [22].
(2)$$E_{adh}=\gamma_{m}+\gamma_{s}-\gamma_{m/s}$$

According to Equation (2), when E_adh_ < 2γ_m_, the initial phase of film growth is island growth. According to the Eadh and γ_m_ values of different metals on a SiO_2_ substrate, gold (Au), silver (Ag) and copper (Cu) all have E_adh_ < 2γ_m_ values, while aluminum has a higher adhesion energy [23]. The growth of these metal films (e.g., Ag, Au, and Cu) typically follows the Volmer–Weber mode, and isolated metallic islands, instead of continuous metallic layers, are formed on the substrate in the early stage of film growth [24]. In addition, from the point of view of dynamics, Bauer points out that islands tend to grow on the thermodynamic model of the case. If the film has a large enough force between the atoms and the substrate to bind the atoms on the surface of the substrate. The diffusion film on the surface of the substrate may also be in accordance with the island growth pattern, showing that the reaction between the film and the substrate will also impact on the thin film growth mode [25]. By observing the surface morphology of Ag films with different thicknesses on a ZnO film surface, Yun verified that Ag grows on the oxide surface in accordance with the island pattern. At the same time, the observation results also show that with the decrease in the core density and cluster density during the growth process, the clustering of film clusters becomes the key factor affecting the morphology of the film [20].

Therefore, the initial growth of gold, silver and copper metal films follows the island growth pattern, and finally, the islands are connected to form a film, resulting in a greater threshold thickness of the silver film. In order to obtain a transparent conductive film with good performance, it is necessary to overcome the island growth mode and reduce the threshold thickness of the film. Therefore, aluminum can be used as the seed layer to reduce the threshold thickness of the silver film and prepare a transparent conductive silver film with good performance. In addition, the infiltration effect of the Al seed layer was further improved by changing its infiltration mode.

### 2.2. Experiment

For the experiments, K9 glass with a diameter of 25 mm and a thickness of 2 mm was selected as the substrate. The substrate samples were cleaned to remove contamination and improve film quality and uniformity. This was achieved with multiple sonicated solvent rinses in acetone and isopropyl alcohol, and drying with pressurized nitrogen (99.99%). The aluminum and silver films were prepared with a high vacuum resistance evaporation coating machine manufactured by Tekono Technology Co., Ltd, Beijing, China. The device has two evaporation boats, but only one evaporation power supply, to be switched to deposit aluminum and silver. During the preparation process, the background vacuum was 5.0 × 10^−3^ Pa. The schematic diagram of the equipment is shown in Figure 2. Deposition rates for the thermal evaporation processes were monitored using quartz crystal oscillators. The film thickness was set in advance, the current was turned on for preheating, and the rotation speed of the substrate was set to 10 r/min. Then, the baffle was opened for the preparation of aluminum and silver. The evaporation current was between 90 A and 120 A, the deposition rate of aluminum was 1 Å/s, and the deposition rate of silver was 3 Å/s. In the preparation process, the quartz crystal vibration film thickness meter of the equipment was used to monitor the film thickness. Before the experiment, the tool factor of the quartz crystal vibration film thickness monitor was calibrated using a MultiMode8 scanning probe microscope (Bruker, Billerica, MA, USA); that is, a thick film was prepared, and the thickness of the step was measured with the scanning probe microscope through a step made of high-temperature tape on the silicon substrate. The ratio between the actual thickness of the experimentally prepared film and the thickness set on the equipment during the experiment is the specific value of the tool factor. After determining the specific value of the tool factor, a series of experiments were carried out.

### 2.3. Thin Film Characterization

After the preparation of a thin silver film infiltrated with an extreme aluminum seed layer by heat evaporation, a series of characterizations of the film were carried out directly to prevent oxidation of the film from affecting its effect. The wavelength range of 400–2500 nm was measured with a U-3501 UV-visible spectrophotometer manufactured by Hitachi; and the surface morphology of the films was measured using a SU8010 cold-field-emission scanning electron microscope (Carl Zeiss AG, Oberkochen, Germany) to compare films of different thicknesses; the images were obtained with in-lens and SE2 detectors. The electrical properties were measured by a ST2558B-F01 film linear four-probe test platform produced by Lattice Electronics Co., Ltd, Suzhou, China.

## 3. Results and Discussion

### 3.1. Preparation of Single Layer Silver Film

Firstly, we prepared a single layer silver film on a K9 glass substrate. When the thickness of the silver film is thinner, the transmittance is higher, and decreases as the thickness of the silver film increases. Although the transmittance of 5 nm and 8 nm silver films is higher, there is no square resistance, and the silver films are not conductive. With increased thickness, the transmittance of 15 nm and 25 nm silver films is low, but the square resistance is good, at around 10 Ωsq ^−1^. The transmittance curves and surface morphology of silver films with different thicknesses are shown in Figure 3. From the transmittance curve, it can be seen that when the silver film is too thin, its sheet resistance value is too high, and conversely, when the silver film is too thick, its transmittance is too low. Therefore, it is necessary to improve the transmittance of the silver film while ensuring good sheet resistance. As can be seen from the SEM image, when the silver film is thin, it is obviously granular. As the thickness increases, the particles gradually become larger, from island film to layer film, and finally form a massive silver film. In other words, the silver film without a seed layer cannot achieve a transparent conductive effect. Therefore, it is necessary to introduce a seed layer to reduce the threshold thickness of the silver film, so that it can achieve higher transmittance and better sheet resistance at a lower thickness, which can meet the requirements of transparent conductive films.

### 3.2. Optimization of Seed Layer Thickness

The thickness range of aluminum seed layer was determined by reading relevant literature and conducting exploratory experiments in the early stage. Aluminum films of 1 nm and 2 nm were prepared on a K9 glass substrate and used as the seed layer of silver film. After plating silver films of different thickness on the seed layer, it was found that the infiltration effect of a 2 nm aluminum seed layer on the silver film was not obvious; that is, compared with the single layer silver film, the performance of the silver film was not greatly improved. Sheet resistance can be measured in silver films with 1 nm Al seed layer, indicating that the silver film is relatively continuous. Then, the thickness around 1 nm of the aluminum seed layer was refined and studied. The following figure shows the sheet resistance curves of the silver film with a 0.8 nm, 1 nm and 1.2 nm aluminum seed layer, respectively, for 8 nm and 10 nm silver film.

Figure 4 shows the sheet resistance diagram with error bar, comparing the influence of different aluminum seed layers on the sheet resistance of silver film [26]. As can be ascertained from Figure 4, the 1 nm-thick aluminum seed layer relative to the 0.8 and 1.2 nm aluminum seed layer has good electrical conductivity, suggesting that silver film with a 1 nm aluminum seed layer is more continuous. In other words, the silver film with 1 nm aluminum seed layer has a better conductivity, and this conclusion can be obtained for both 8 nm and 10 nm silver films. To further verify the infiltration effect of the 1 nm aluminum seed layer on the silver film, a cold-field emission scanning electron microscopy (SEM) test was performed on the silver film with different thickness of the aluminum seed layer. The test results are shown in the following figure (Figure 5). The effect of the 1 nm aluminum seed layer on the silver film can be clearly seen in the figure.

### 3.3. Optimization of Silver Film Thickness

After determining the optimal thickness of the aluminum seed layer, silver films with different thickness were plated on a 1 nm aluminum seed layer by the same process to study the optical and electrical properties. Figure 6a shows the transmittance curves of silver films with different thicknesses on a 1 nm aluminum seed layer in the band of 400–2500 nm measured by spectrophotometer, and the transmittance in the figure does not take into account backside reflection. Figure 6b shows the sheet resistance error bars of silver films with different thicknesses on a 1 nm aluminum seed layer measured by linear four-probe test platform [26].

The dispersion characteristics of very thin silver films vary with film thickness, while the dispersion characteristics of continuous Ag films are very small [27]. It can be seen in Figure 6a that when the silver films are thin, the transmittance of the silver films at 7 nm is relatively high, and this increases with the change in wavelength, which is mainly caused by the dispersion. When the thickness of silver film is above 8 nm, the transmittance decreases as the silver film thickness increases. The electrical characteristics are tested by the thin-film linear four-probe test bench. The maximum resistance value that can be tested by this instrument is 4.48 × 10^36^ Ω sq^−^^1^, and the sheet resistance value of the single-layer 10 nm silver film tested is 4.48 × 10^36^ Ωsq ^−1^; that is, the conductivity is almost zero. It can be seen from the sheet resistance curve that the sheet resistance of the silver film with an aluminum seed layer is significantly improved with the increase in the thickness of the silver film. The transmittance of the 7 nm-thick silver film is higher; that is, the film is not continuous enough, resulting in a larger sheet resistance. When the thickness of the silver film reaches 10 nm, the sheet resistance can reach about 13 Ω sq^−^^1^.

Figure 7 shows the surface morphology images of silver films with different thickness on a 1 nm aluminum seed layer measured by cold-field emission scanning electron microscopy. It can be clearly observed from the figure that the silver films with thicknesses less than 10 nm have obvious furrows and do not form relatively continuous films. When the thickness reaches 10 nm, the number of furrows decreases, and the furrows begin to connect to form a relatively continuous film. With the increase in the silver film thickness, the silver film larger than 10 nm has become significantly continuous, and the sheet resistance decreases with the increase in the silver film thickness, but the transmittance will be affected. By comparing the 10 nm silver film with a 1 nm aluminum seed layer as an infiltration layer and the 10 nm silver film without an aluminum seed layer, it can be seen that the folds of the silver film without an aluminum seed layer are significantly increased, while the silver film with a seed layer is more continuous, and the infiltration effect of the aluminum seed layer can be compared.

### 3.4. Optimization of Preparation Method for Aluminum Seed Layer

From the previous experimental results, it can be seen that the introduction of an aluminum seed layer can change the growth mode of silver film, reduce the threshold thickness of silver film, and improve the surface morphology of silver film, so that the silver film has a low sheet resistance on the premise of maintaining a high transmittance. The introduction of an aluminum seed layer has a certain infiltration effect on the silver film, and although the conductivity is good, the transmittance needs to be improved. Then, we proposed a new method to prepare the seed layer; that is, on the basis of the 1 nm aluminum seed layer obtained in the previous example, the aluminum seed layer and silver film were deposited layer by layer to approximately achieve a state of mixed evaporation. This can make the seed layer play the role of infiltration better, as shown in Figure 8.

As can be seen in Figure 8, the 1 nm aluminum seed layer was deposited into two layers. First, the 0.5 nm aluminum seed layer was deposited on the glass substrate, then the 0.5 nm silver film was deposited, then the 0.5 nm aluminum seed layer was deposited, and finally, silver film of different thickness was deposited on this base. Previously, we also tried to divide the 1 nm aluminum seed layer into three or four layers for deposition, but the experimental results show that the deposition effect of two layers is more obvious, and the process of two layers is relatively simple and convenient.

The transmittance curves of silver films with different thicknesses, after the aluminum seed layer was deposited separately, are shown below, where the sheet resistance values are also indicated. It can also be seen from the transmittance curve in Figure 9 that the transmittance gradually decreases with the increase in the thickness of the silver film. However, compared with the silver film with the 1 nm aluminum seed layer directly deposited before, the effect of separating the aluminum seed layer is more obvious, and the highest transmittance can reach about 60%, with the sheet resistance being less than 100 Ωsq ^−1^. The transmittance of very thin silver films prepared by this method is improved, but the sheet resistance is not much reduced. This conclusion can be further verified by the SEM image.

As shown in Figure 10, the aluminum seed layer and the silver film are alternately deposited. The thickness of the last layer of silver film is constantly changing. It can be seen from the figure that as the thickness of the last layer of silver film increases, the silver film is obviously more continuous. Although there are obvious gullies, it can be seen that the silver film is obviously connected to one piece. This is also the reason for the high transmittance and low sheet resistance.

## 4. Conclusions

In this work, through measurement of the transmittance, sheet resistance and SEM imaging of the experimentally prepared ultra-thin silver conductive film, it can be concluded that the silver film with the aluminum seed layer is significantly more continuous than the silver film without aluminum seed layer; that is, the introduction of an aluminum seed layer can make the silver film continuous at a lower thickness. Furthermore, the continuous threshold thickness of the silver film is reduced. Aluminum has a better affinity with glass substrates than silver. The optimal thickness of aluminum as a seed layer is 1 nm, and the transmittance of 10 nm-thick silver film can reach about 40% in the band of 400–2500 nm, and the minimum square resistance can reach 13 Ωsq ^−1^. After that, by changing the infiltration mode of the aluminum seed layer and depositing it alternately with silver film, thinner transparent conductive silver film can be obtained. The transmittance of 5 nm silver film can reach about 60% in the band of 400–2500 nm, and the sheet resistance value can reach 90 Ω sq^−^^1^ at the lowest. The results showed that alternating deposition could play a better role in the infiltration of the aluminum seed layer, and the infiltration effect was better than that of using 1 nm aluminum directly as the infiltration layer.

## Figures and Tables

**Figure 1 nanomaterials-12-03540-f001:**
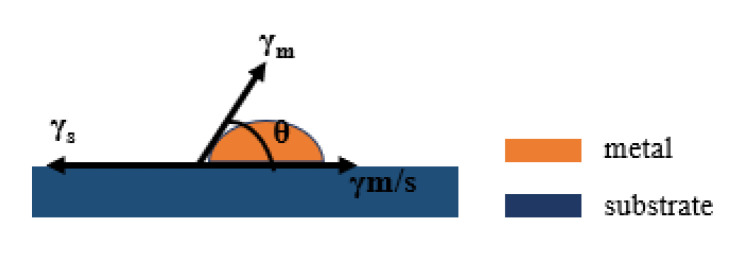
Schematic diagram of surface energy during metal film deposition.

**Figure 2 nanomaterials-12-03540-f002:**
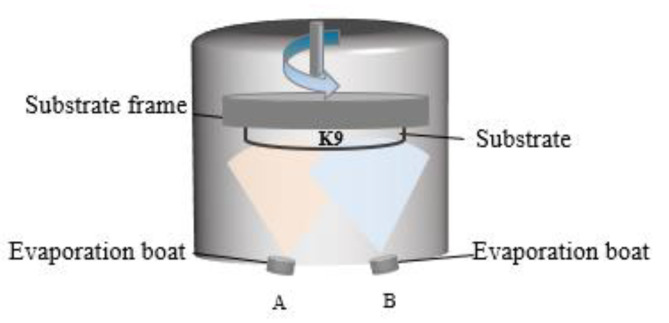
Schematic diagram of thermal evaporation of resistance.

**Figure 3 nanomaterials-12-03540-f003:**
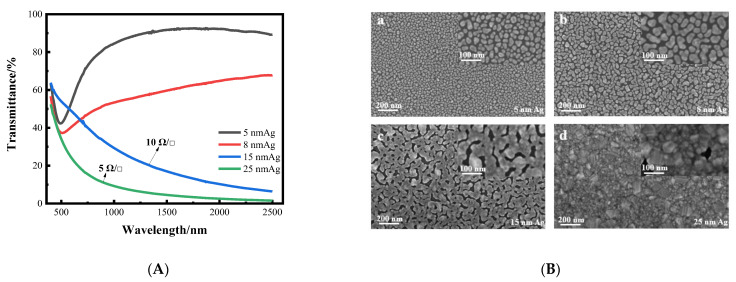
(**A**) Curves of transmittance of silver films with different thickness varying with wavelength. (**B**) SEM images of silver films with different thickness on a K9 substrate (**a**–**d**).

**Figure 4 nanomaterials-12-03540-f004:**
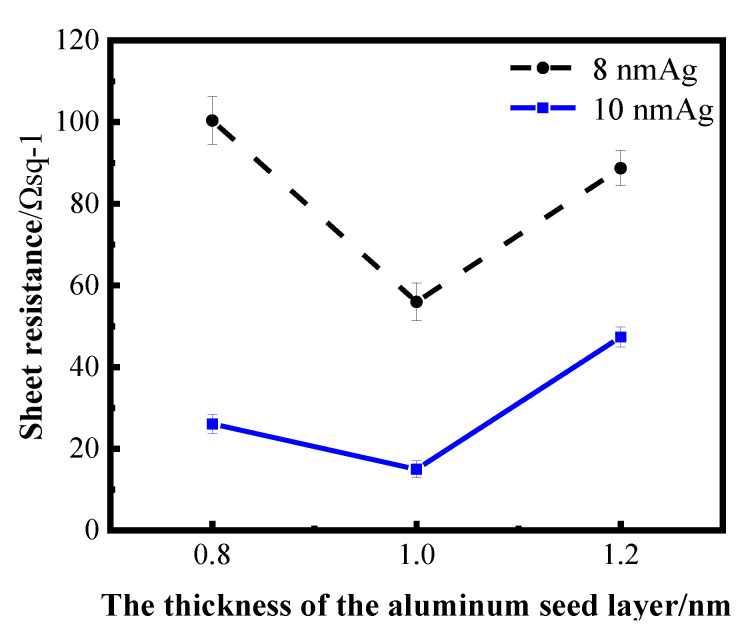
Curves of sheet resistance of silver film with different thickness on an aluminum seed layer.

**Figure 5 nanomaterials-12-03540-f005:**
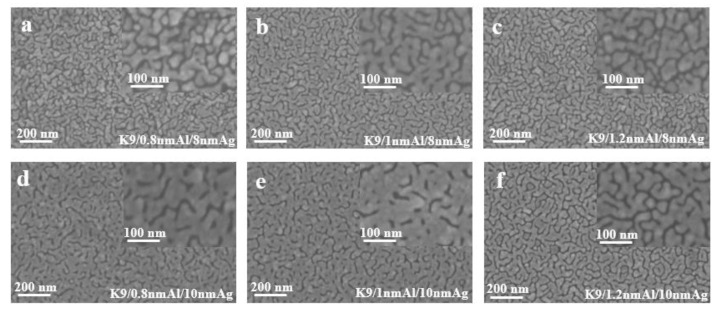
SEM images of silver films with aluminum seed layer of different thickness (**a**–**f**).

**Figure 6 nanomaterials-12-03540-f006:**
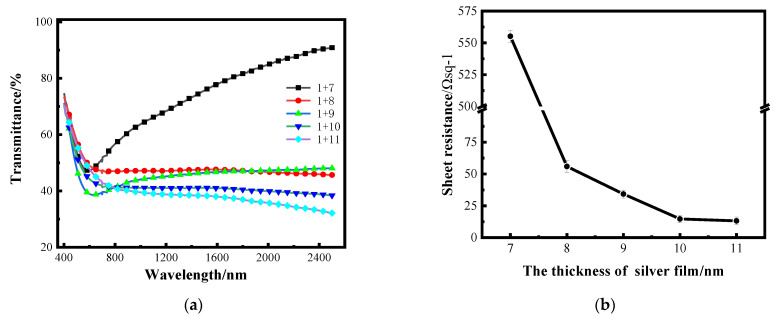
(**a**) Transmittance curves of silver films with different thickness on the surface of a 1 nm aluminum seed layer; (**b**) the sheet resistance curve of silver film with different thickness on an aluminum seed layer.

**Figure 7 nanomaterials-12-03540-f007:**
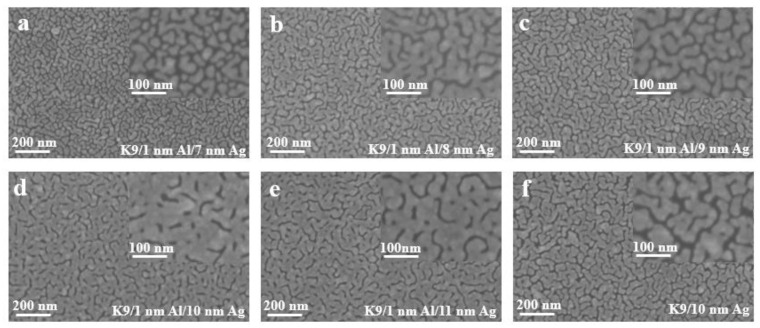
(**a**–**e**) SEM images of silver film with different thickness on the surface of a 1 nm aluminum seed layer. (**f**) SEM images of silver film with 10 nm on K9 substrate.

**Figure 8 nanomaterials-12-03540-f008:**
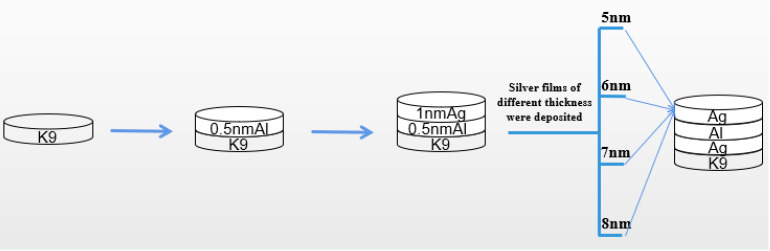
Depiction of three different structures of silver films.

**Figure 9 nanomaterials-12-03540-f009:**
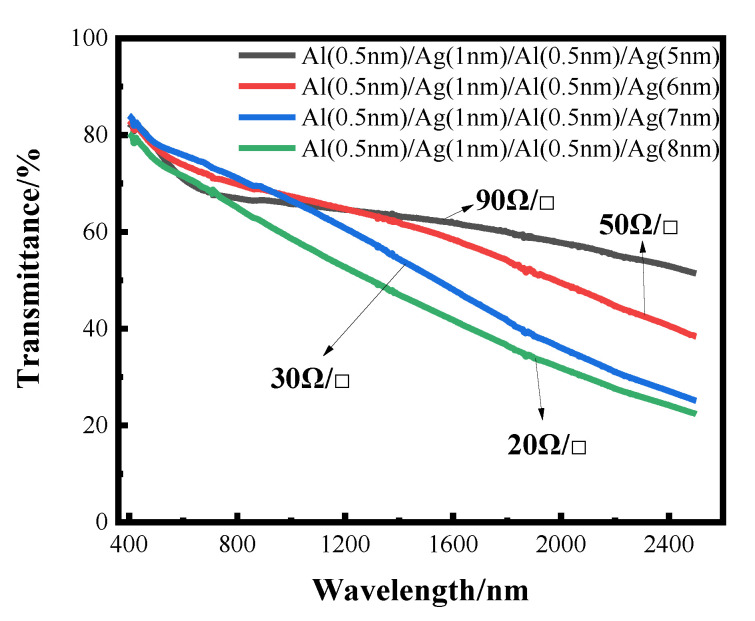
The transmittance curve of transmittance as a function of wavelength for alternating deposition of Al seed layer and Ag film.

**Figure 10 nanomaterials-12-03540-f010:**
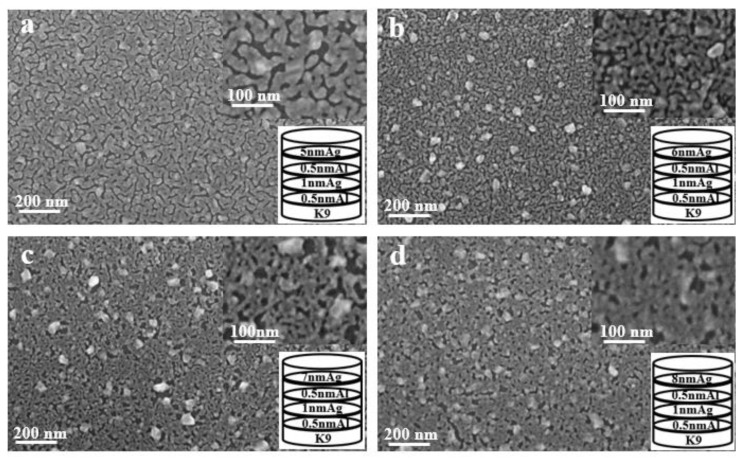
SEM images of alternating deposition of aluminum seed layer and silver film (**a**–**d**).

## Data Availability

The data supporting the findings of this study are available by reasonable request to the first author.

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
