# Peer review of "Study on Ultrathin Silver Film Transparent Electrodes Based on Aluminum Seed Layers with Different Structures"

_nanomaterials, 2022, doi:10.3390/nano12193540_

Round 1

Reviewer 1 Report

The authors present an interesting experimental study of fabrication of ultra-thin silver transparent electrode based on aluminum seed layer.

This paper is written in a good way pointing out very critical things. However, I suggest that the author could modify or add a few things below.

1.     Please check the English used in the article. Some sentences or phrases seem to be incorrect. Please see the comments in the attached pdf file.

2.     Could the authors estimate the roughness of the proposed electrode?

3.     Could the authors measure the hydrophobicity of the prepared films?

4.     Please rewrite the inscription of Figures 3, 4, 6, 8.

Author Response

Dear reviewer

We thank you very much for giving us an opportunity to revise our manuscript. Thank you very much for your email of september 15, 2022 concerning our manuscript (nanomaterials-1924734 entitled “Demonstration of ultra-thin silver film transparent electrode based on aluminum seed layer"). We have tried our best to revise our manuscript according to the comments. Attached please find the revised version, which we would like to submit for your kind consideration. We hope that the revised manuscript resolves the problems pointed out by the reviewers and meets your journal’s standards. In response to your letter, we would like to reply to the comments as detailed below:

Yours sincerely,

Mr. Dong Li

  1. Please check the English used in the article. Some sentences or phrases seem to be incorrect. Please see the comments in the attached pdf file.

      Thank you for the reviewer’s suggestion for this manuscript, I have corrected        the incorrect sentences and phrases in the PDF file you attached, and they're        highlighted in red.

  1. Could the authors estimate the roughness of the proposed electrode?

The RMS of the glass substrate used in the experimental research in the manuscript is 1-2nm. If the silver film is a continuous layered film, the roughness should be almost the same as that of the glass substrate. However, it can be seen from the SEM image that the silver film is not completely continuous, so the RMS of the electrode is about 3-4nm.

  1. Could the authors measure the hydrophobicity of the prepared films?

The ultra-thin silver film studied in this manuscript is mainly to study its optical and electrical properties. The main focus is on how to ensure better conductivity while improving the transmittance of the silver film. The hydrophobicity of the prepared films has not been considered at present, and the transparent electrode studied in the manuscript will be applied to the device and will not be exposed to air.

  1. Please rewrite the inscription of Figures 3, 4, 6, 8.

Based on the reviewer’s suggestion, I have rewrited the inscription of Figures 3, 4, 6, 8.

Figure 3. (a) Curves of transmittance of silver films with different thickness varying with wavelength. (b) SEM images of silver films with different thickness on K9 substrate.

Figure 4. Curves of sheet resistance of silver film with different thickness on aluminum seed layer.

Figure 6. (a,b,c,d,e)SEM imanges of silver film with different thickness on the surface of 1nm aluminum seed layer. (f) SEM imanges of silver film with 10 nm on K9 substrate.

Figure 8.(a) The transmittance curve of transmittance as a function of wavelength for alternating deposition of Al seed layer and Ag film; (b) SEM imanges of alternating deposition of aluminum seed layer and silver film;

Reviewer 2 Report

I have carefully reviewed the paper entitled: "Demonstration of ultra-thin silver film transparent electrode based on aluminum seed layer" Dong Li, Yongqiang Pan, Huan Liu, Yan Zhang, Zhiqi Zheng and Fengyi Zhang. This contribution is interesting from the point of view to obtain another kind of conducting substrate, which can be cheaper than conventional FTO and ITO, with better performance to fabricate optoelectronic and photovoltaic devices.

I have some questions related to this work:

1. What is the reason to obtain the lowest sheet resistance at 1 nm Al seed layer, even the Ag thickness is different?

2. Is there any results of these substrates in the fabrication and performance of optoelectronic and photovoltaic devices?

3. Through XPS, the chemical environment and composition can be established in order to observe the surface of the conducting substrates. This fact can influence the final quality of the deposited material.

Author Response

Dear reviewer

We thank you very much for giving us an opportunity to revise our manuscript. Thank you very much for your email of september 15, 2022 concerning our manuscript (nanomaterials-1924734 entitled “Demonstration of ultra-thin silver film transparent electrode based on aluminum seed layer"). We have tried our best to revise our manuscript according to the comments. Attached please find the revised version, which we would like to submit for your kind consideration. We hope that the revised manuscript resolves the problems pointed out by the reviewers and meets your journal’s standards. In response to your letter, we would like to reply to the comments as detailed below:

Yours sincerely,

Mr. Dong Li

  1. What is the reason to obtain the lowest sheet resistance at 1 nm Al seed layer, even the Ag thickness is different?

The 1nm aluminum seed layer is the best thickness of the infiltrated silver film obtained through multiple experiments. It can be seen from the preliminary exploration experiments that the best thickness of the aluminum seed layer is 1nm for both the 8nm and 10nm silver films1nm.

  1. Is there any results of these substrates in the fabrication and performance of optoelectronic and photovoltaic devices?

At present, the transparent conductive silver film obtained is still in the exploration stage. Experiments have proved that the transparent conductive silver film with lower thickness can be obtained by the introduction of aluminum seed layer,and it has not been specifically applied to photoelectric and photovoltaic devices.So,there is no results about these substrates in the fabrication and performance of optoelectronic and photovoltaic devices.

  1. Through XPS, the chemical environment and composition can be established in order to observe the surface of the conducting substrates. This fact can influence the final quality of the deposited material.

Before the preparation of transparent conductive silver film, the glass substrate was cleaned by ultrasonic to reduce the influence of the glass substrate on the transparent conductive silver film. When preparing the transparent conductive silver film, the pressure in the vacuum chamber is 5×10-3 Pa. During the deposition process, the aluminum seed layer is deposited first, and then the silver film is deposited. No other components be produced during the preparation process. After the preparation of transparent conductive film, the transmittance, square resistance and SEM are directly tested to ensure that the deposited transparent conductive film is not oxidized as much as possible.

Reviewer 3 Report

Manuscript ID: nanomaterials-1924734

Title: Demonstration of ultra-thin silver film transparent electrode based on aluminum seed layer

This manuscript reports on a method to change the growth kinetics of thermally evaporated films of silver(Ag). A seeding layer (1 nm) of aluminum (Al) was used as an infiltrated layer between Ag films to obtain an Ag layer surface with better smooth surface morphology and higher electrical conductivity. The proposed methodology was proven to decrease the threshold thickness of the silver film. This study would be contributed to the development of flexible transparent electrodes with improved properties that are of very relevance in many technological applications. Nevertheless, I think that the scientific content of the paper could be improved by the authors and some issues should be clarified before considering this paper for publication:

1) The authors claimed that Ag could be used as an alternative electrode material to the widely used transparent conductive oxides. The authors should elaborate by listing the advantages/disadvantages of Ag vs. ITO in terms of optical and electrical properties. What is the abundance of Ag on earth? Is it much higher than ITO?

2) In my opinion, the manuscript would benefit from more comprehensive literature data. Only 20 references are provided for the entire text. For instance: “It is well known that Ag grows in the Volmer-Weber mode on glass substrates” – References should be included to support the affirmation.

3) The authors describe the model of thin metal film growth according to the classic theory of the nucleation and growth of thin films. Again, more literature citations will be helpful, and they could help the authors to improve the description of the nucleation and growth of thin films fabricated by vapor deposition methods (e.g. Venables, J.A.; Spiller, G.D.T.; Hanbucken, M. Nucleation and Growth of Thin Films. Rep. Progr. Phys. 1984, 47, 399–459 & Venables, J.A.; Spiller, G.D.T. Nucleation and Growth of Thin Films. In Surface Mobilities on Solid Materials; Springer: Boston, MA, USA, 1983; 341–404 & Ratsch, C.; Venables, J.A. Nucleation Theory and the Early Stages of Thin Film Growth. J. Vac. Sci. Technol. A Vacuum Surfaces Films 2003, 21, S96–S109 & Costa, J.C.S.; Coelho, A.F.S.M.G.; Mendes, A.; Santos, L.M.N.B.F. Nucleation and Growth of Microdroplets of Ionic Liquids Deposited by Physical Vapor Method onto Different Surfaces. Appl. Surf. Sci. 2018, 428, 242–24). Young´s equation was presented to describe the surface energy during metal film deposition. The authors are advised of the fact that Young's equation assumes that the system is macroscopic and very stable. The interpretations should be taken with some care for systems at micro- or nanoscale levels. Additional details should be included. In addition, the film morphology also depends on the type of surface roughness. Additional comments should be provided.

4) Details of the SEM characterization were not presented. The images were obtained with SE or BSE detectors? SEM images were taken how long after the deposition? The coalescence of Ag clusters would not be dependent on the time after the deposition?

5) Figure 5 would benefit if the SEM images of Ag films with an Al infiltration layer were compared with SEM images of Ag films deposited directly onto glass.

6) What is the real effect of the Al seeding layer on the nucleation and growth of Ag films? The authors could elaborate by making conclusions on the different adhesion/affinity/surface energy of the Ag clusters onto Al or glass. Do the authors have a possible explanation for a better result for 1 nm of Al instead of 0.8 nm or 1.2 nm? Possible explanations for their finding would improve the scientific quality of the paper.

7) The authors used different evaporation rates to deposit Al and Ag. Why? Did the authors test the effect of the deposition rate on the film morphology?

8) In Figure 5a the authors can make clear what means 1+7, 1+8, 1+9… for instance using the legend Ag(7nm)/Al(1nm) instead of 1+7. This would be more useful for the readers to better visualize/understand the results. In this figure, the transmittance curve of the silver film without Al should be included for comparison.

9) The quality of Figure 7 should be improved. Details could be added to the scheme such as the thickness of each layer.

10) In Figure 8a, what means 0.5+1+0.5+5 ? These values are different from those presented in Figure 8b (SEM images): 0.5+0.5+0.5… The schemes inside the SEM images are not perceptible. Please improve the quality/size of the schemes.

11) In the abstract and conclusion sections, the authors are requested to elaborate. What is the role of the Al seeding layer and what is the main reason for improving the morphological/electrical characteristics? Is there a preferential affinity of Ag to the Al rather than to the glass? Will have the Ag clusters lower mobility onto the Al surface due to a lower interfacial tension which contributes to an improved wettability? Does Ag also grow in the Volmer-Weber mode on Al/glass substrates? What are the advantages of making multilayer depositions with Ag and Al alternately deposited? Would be a good idea to deposit both metals simultaneously?

Best regards.

Author Response

Dear reviewer

We thank you very much for giving us an opportunity to revise our manuscript. Thank you very much for your email of september 15, 2022 concerning our manuscript (nanomaterials-1924734 entitled “Demonstration of ultra-thin silver film transparent electrode based on aluminum seed layer"). We have tried our best to revise our manuscript according to the comments. Attached please find the revised version, which we would like to submit for your kind consideration. We hope that the revised manuscript resolves the problems pointed out by the reviewers and meets your journal’s standards. In response to your letter, we would like to reply to the comments as detailed below:

Yours sincerely,

Mr. Dong Li

1) The authors claimed that Ag could be used as an alternative electrode material to the widely used transparent conductive oxides. The authors should elaborate by listing the advantages/disadvantages of Ag vs. ITO in terms of optical and electrical properties. What is the abundance of Ag on earth? Is it much higher than ITO?

Based on the reviewer’s suggestion, the advantages/disadvantages of Ag has been discussed in the paper.

Indium tin oxide (ITO) is the conventional selection and most widely used for the transparent electrode because of its high visible transmittance and electrical conductivity. However, the low abundance of the indium element on earth is a limiting factor of this material. In addition, its applications in emerging flexible optoelectronic devices are significantly hindered by both the poor mechanical flexibility and the high annealing temperature needed to reduce its resistivity. Recently, several other transparent conductive materials, which have been developed to address these issues. For instance, doped metal oxides, thin metals, conducting polymers, and nanomaterials (including carbon nanotubes, graphene, and metal nanowires)[6], etc, have gradually become effective substitutes for ITO film. Metal thin film is an ideal material for transparent electrode because of its simple preparation process, low cost, excellent mechanical flexibility and uniform photoelectric property.And compared with the ITO film, the transmittance of the metal silver film is relatively poor, but the silver film has good conductivity and flexibility, and the silver resource is richer than the indium resource[7].

[7]Zhong,T.;Han,Y.;Qing,P.;Lin,J,G.; et al. Atomistic Insights into Aluminum Doping Effect on Surface Roughness of Deposited Ultra-Thin Silver Films. Nanomaterials2021,1,11,158.

2) In my opinion, the manuscript would benefit from more comprehensive literature data. Only 20 references are provided for the entire text. For instance: “It is well known that Ag grows in the Volmer-Weber mode on glass substrates” – References should be included to support the affirmation.

Based on the reviewer’s suggestion, more comprehensive literature have been listed in the manuscript.

3) The authors describe the model of thin metal film growth according to the classic theory of the nucleation and growth of thin films. Again, more literature citations will be helpful, and they could help the authors to improve the description of the nucleation and growth of thin films fabricated by vapor deposition methods (e.g. Venables, J.A.; Spiller, G.D.T.; Hanbucken, M. Nucleation and Growth of Thin Films. Rep. Progr. Phys. 1984, 47, 399–459 & Venables, J.A.; Spiller, G.D.T. Nucleation and Growth of Thin Films. In Surface Mobilities on Solid Materials; Springer: Boston, MA, USA, 1983; 341–404 & Ratsch, C.; Venables, J.A. Nucleation Theory and the Early Stages of Thin Film Growth. J. Vac. Sci. Technol. A Vacuum Surfaces Films 2003, 21, S96–S109 & Costa, J.C.S.; Coelho, A.F.S.M.G.; Mendes, A.; Santos, L.M.N.B.F. Nucleation and Growth of Microdroplets of Ionic Liquids Deposited by Physical Vapor Method onto Different Surfaces. Appl. Surf. Sci. 2018, 428, 242–24). Young´s equation was presented to describe the surface energy during metal film deposition. The authors are advised of the fact that Young's equation assumes that the system is macroscopic and very stable. The interpretations should be taken with some care for systems at micro- or nanoscale levels. Additional details should be included. In addition, the film morphology also depends on the type of surface roughness. Additional comments should be provided.

Thank you for the reviewer’s suggestion , I added the following to the manuscript.

According to Equation (2), when Eadh<2γm ,the initial phase of film growth is island growth. According to the Eadh and γm values of different metals on SiO2 substrate, gold (Au), silver (Ag) and copper (Cu) all have Eadh<2γm values, while aluminum has a higher adhesion energy[25].The growth of these metal films (e.g.Ag, Au, and Cu) typically follows the Volmer–Weber mode and isolated metallic islands, instead of continuous metalliclayers, are formed on the substrate in the early stage of film growth[26]. In addition, Bauer and also from the point of view of dynamics, points out that islands tend to grow on the thermodynamic model of the case, if the film has a large enough force between atoms and substrate to bind atoms on the surface of the substrate, the diffusion film on the surface of the substrate may also be in accordance with the island growth pattern, show that between film and substrate reaction will also impact on thin film growth mode[27]. By observing the surface morphology of Ag films with different thicknesses on ZnO film surface, Yun verified that Ag grows on oxide surface in accordance with the island pattern. At the same time, the observation results also show that with the decrease of the core density and cluster density during the growth process, the clustering of film clusters becomes the key factor affecting the morphology of the film[28].Therefore, the initial growth of gold, silver and copper metal films follow the island growth pattern, and finally the island is connected to form a film, resulting in a larger threshold thickness of silver film. In order to obtain the transparent conductive film with good performance, it is necessary to overcome the island growth mode and reduce the threshold thickness of the film. Therefore, aluminum can be used as the seed layer to reduce the threshold thickness of the silver film and prepare the transparent conductive silver film with good performance.

[27]Bauer E.Zeitschrift für Kristallographie-Crystalline Materials,1958,110 ( 1-6) ,395.

[28]Yun,J.;Ultrathin Metal films for Transparent Electrodes of Flexible Optoelectronic DevicesAdvanced Functional Materi-als,2017,27(18)1606641.

4) Details of the SEM characterization were not presented. The images were obtained with SE or BSE detectors? SEM images were taken how long after the deposition? The coalescence of Ag clusters would not be dependent on the time after the deposition?

Thank you for the suggestion, the details of the SEM characterization have been added in the manuscript. After experiment, SEM characterization of the deposited silver film was carried out directly, so the effect of time on the coalescence of Ag clusters was not considered or studied. And the details of the SEM characterization were presented as follows.

And the surface morphology of the films was measured by SU8010 cold field emission scanning electron microscope (SEM) to compare the films of different thickness, and the images were obtained with  SE2 detectors.

5) Figure 5 would benefit if the SEM images of Ag films with an Al infiltration layer were compared with SEM images of Ag films deposited directly onto glass.

Figure 5 is to further verify the conclusion obtained in Figure 4, and to further determine the optimal infiltration thickness of aluminum seed layer, so the surface morphology of silver film deposited on different aluminum seed layer thickness is compared.

6) What is the real effect of the Al seeding layer on the nucleation and growth of Ag films? The authors could elaborate by making conclusions on the different adhesion/affinity/surface energy of the Ag clusters onto Al or glass. Do the authors have a possible explanation for a better result for 1 nm of Al instead of 0.8 nm or 1.2 nm? Possible explanations for their finding would improve the scientific quality of the paper.

The real effect of the aluminum seed layer is to improve the adhesion between the silver film and the glass substrate. Through the aluminum seed layer, the silver film is connected as much as possible to ensure a high pass rate and electrical conductivity. We are specifically characterized by electrical conductivity, because the same thickness of silver film, without aluminum seed layer is not conductive.

The possible explanation for a better result for 1 nm of Al instead of 0.8 nm or 1.2 nm, I think the possible reason is that the thickness of 1nm is just enough to make the silver film continuous and ensure a better transmittance. Too thick aluminum seed layer may cause the silver film to continue to grow on the island aluminum seed layer, which cannot guarantee high transmittance and electrical conductivity. If the aluminum seed layer is too thin, the possible reason is that the aluminum seed layer does not play the corresponding role.

7) The authors used different evaporation rates to deposit Al and Ag. Why? Did the authors test the effect of the deposition rate on the film morphology?

The low deposition rate of aluminum seed layer is mainly due to the thin aluminum seed layer. In order to better control the thickness of aluminum seed layer, the deposition rate of aluminum seed layer is slow. The influence of different deposition rates (1 Å/s,3 Å/s and 5 Å/s) on the transmittance and sheet resistance of silver films has been studied before the deposition of silver films. The experimental results show that the deposition rate has almost no effect on the transmittance and square resistance of silver films. Finally, an appropriate deposition rate was chosen.

8) In Figure 5a the authors can make clear what means 1+7, 1+8, 1+9… for instance using the legend Ag(7nm)/Al(1nm) instead of 1+7. This would be more useful for the readers to better visualize/understand the results. In this figure, the transmittance curve of the silver film without Al should be included for comparison.

Thank you for the suggestion, I have modified 1+7,1+8,1+9…… in Figure 5a to make it clearer. And the main purpose of this figure is to illustrate the transmittance of silver films of different thickness on the 1nm aluminum seed layer. The transmittance of single layer silver film has been explained in Section 3.1. Compared with the transmittance of single-layer silver film, the transmittance of silver film with aluminum seed layer is lower, so this part is not compared again.

9) The quality of Figure 7 should be improved. Details could be added to the scheme such as the thickness of each layer.

Thank you for the reviewer’s suggestion, Fig. 7 is an idea to express the separate deposition of aluminum seed layers. According to your suggestion, I have improved the quality of Fig. 7 and added the thickness of each layer.

10) In Figure 8a, what means 0.5+1+0.5+5 ? These values are different from those presented in Figure 8b (SEM images): 0.5+0.5+0.5… The schemes inside the SEM images are not perceptible. Please improve the quality/size of the schemes.

Thank you for the suggestion, I am sorry to make a mistake. I have unified the thickness representation in Figs. 8a and 8b, and I have improved the quality/size of the schematics.

11) In the abstract and conclusion sections, the authors are requested to elaborate. What is the role of the Al seeding layer and what is the main reason for improving the morphological/electrical characteristics? Is there a preferential affinity of Ag to the Al rather than to the glass? Will have the Ag clusters lower mobility onto the Al surface due to a lower interfacial tension which contributes to an improved wettability? Does Ag also grow in the Volmer-Weber mode on Al/glass substrates? What are the advantages of making multilayer depositions with Ag and Al alternately deposited? Would be a good idea to deposit both metals simultaneously?

Thank you for the reviewer’s suggestion, I added the following to the manuscript.

In this work, through the measurement of the transmittance, sheet resistance and SEM image of the experimentally prepared ultra-thin silver conductive film, it can be concluded that the silver film with aluminum seed layer is significantly more continuous than the silver film without aluminum seed layer, that is, the introduction of aluminum seed layer can make the silver film continuous at a lower thickness, That is, the continuous threshold thickness of the silver film is reduced. Aluminum has a better affinity with glass substrates than silver.The optimal thickness of aluminum as seed layer is 1 nm, and the transmittance of 10 nm thick silver film can reach about 40% in the band of 400−2500 nm, and the minimum sheet resistance can reach 13 Ωsq-1. After that, by changing the infiltration mode of aluminum seed layer and depositing it alternately with silver film, thinner transparent conductive silver film can be obtained. The transmittance of 5 nm silver film can reach about 60% in the band of 400−2500 nm, and the square resistance value can reach 90 Ωsq-1at the lowest. The results showed that alternating deposition could play a better role in the infiltration of aluminum seed layer, and the infiltration effect was better than that of using 1nm aluminum directly as the infiltration layer.

Round 2

Reviewer 1 Report

The authors present an interesting experimental study of fabrication of ultra-thin silver transparent electrode based on aluminum seed layer.

This paper is written in a good way pointing out very critical things, and the manuscript has been revised well according to the proposed comments. Therefore, I don't have any further comments/suggestions. 

Author Response

Dear reviewer

We thank you very much for giving us an opportunity to revise our manuscript. Thank you very much for your email of october1, 2022 concerning our manuscript (nanomaterials-1924734 entitled “Demonstration of ultra-thin silver film transparent electrode based on aluminum seed layer").

Yours sincerely,

Mr. Dong Li

Reviewer 3 Report

Manuscript ID: nanomaterials-1924734

Title: Demonstration of ultra-thin silver film transparent electrode based on aluminum seed layer

Dear Editors and Reviewers,

The authors improved the quality of presentation of their results and tried to answer to all the questions/comments in in comprehensive manner. In my opinion, I believe the manuscript might be suitable for publication. On the other hand, the authors should improve the English quality used in the article. 

Author Response

Dear reviewer

We thank you very much for giving us an opportunity to revise our manuscript. Thank you very much for your email of october1, 2022 concerning our manuscript (nanomaterials-1924734 entitled “Demonstration of ultra-thin silver film transparent electrode based on aluminum seed layer"). We have tried our best to revise our manuscript according to the comments. Attached please find the revised version, which we would like to submit for your kind consideration. We hope that the revised manuscript resolves the problems pointed out by the reviewers and meets your journal’s standards. In response to your letter, we would like to reply to the comments as detailed below:

I want to change the title to “Study on Ultrathin Silver Film Transparent Electrodes Based on Aluminum Seed Layers with Different Structures”.

And I have improved the English quality used in the article. 

Yours sincerely,

Mr. Dong Li
